# Psychosocial Stress Hastens Disease Progression and Sudden Death in Mice with Arrhythmogenic Cardiomyopathy

**DOI:** 10.3390/jcm9123804

**Published:** 2020-11-24

**Authors:** Jacopo Agrimi, Arianna Scalco, Julia Agafonova, Larry Williams III, Nainika Pansari, Gizem Keceli, Seungho Jun, Nadan Wang, Francesca Mastorci, Crystal Tichnell, Brittney Murray, Cynthia A. James, Hugh Calkins, Tania Zaglia, Nazareno Paolocci, Stephen P. Chelko

**Affiliations:** 1Department of Medicine, Johns Hopkins University School of Medicine, Baltimore, MD 21205, USA; jagrimi1@jhmi.edu (J.A.); juliaagafonova7@gmail.com (J.A.); lwill206@jhu.edu (L.W.III); npansar2@jhu.edu (N.P.); gkeceli1@jhu.edu (G.K.); sjun7@jhmi.edu (S.J.); nwang31@jhmi.edu (N.W.); ctichne1@jhmi.edu (C.T.); bdye1@jhmi.edu (B.M.); cjames7@jhmi.edu (C.A.J.); hcalkins@jhmi.edu (H.C.); 2Department of Biomedical Sciences, University of Padova, 35122 Padova, Italy; arianna.scalco@gmail.com (A.S.); tania.zaglia@unipd.it (T.Z.); 3Clinical Physiology Institute, CNR, 56124 Pisa, Italy; mastorcif@ifc.cnr.it; 4Department of Biomedical Sciences, Florida State University College of Medicine, Tallahassee, FL 32306, USA

**Keywords:** arrhythmia, psychosocial stress, desmosomal variants, Desmoglein-2, resident-intruder, anxiety, corticosterone

## Abstract

Physiological stressors, such as exercise, can precipitate sudden cardiac death or heart failure progression in patients with arrhythmogenic cardiomyopathy (ACM). Yet, whether and to what extent a highly prevalent and more elusive environmental factor, such as psychosocial stress (PSS), can also increase ACM disease progression is unexplored. Here, we first quantified perceived stress levels in patients with ACM and found these levels correlated with the extent of arrhythmias and cardiac dysfunction. To determine whether the observed correlation is due to causation, we inflicted PSS-via the resident-intruder (RI) paradigm—upon Desmoglein-2 mutant mice, a vigorously used mammalian model of ACM. We found that ACM mice succumbed to abnormally high in-trial, PSS mortality. Conversely, no sudden deaths occurred in wildtype (WT) counterparts. Desmoglein-2 mice that survived RI challenge manifested markedly worse cardiac dysfunction and remodeling, namely apoptosis and fibrosis. Furthermore, WT and ACM mice displayed similar behavior at baseline, but Desmoglein-2 mice exhibited heightened anxiety following RI-induced PSS. This outcome correlated with the worsening of cardiac phenotypes. Our mouse model demonstrates that in ACM-like subjects, PSS is incisive enough to deteriorate cardiac structure and function per se, i.e., in the absence of any pre-existing anxious behavior. Hence, PSS may represent a previously underappreciated risk factor in ACM disease penetrance.

## 1. Introduction

Stress is a formulaic term often used to define any condition of mental, emotional, or physical strain and/or tension, challenging the organismal mechanisms of adaptation and self-survival. Yet, exercise, for instance (i.e., physiological stress), is quite different from demanding psychosocial conditions: the former can alleviate individuals from anxiety [1], while the latter fuels it [2,3]. In addition, this striking dissimilarity subtends the action of distinct hormonal and non-hormonal factors.

Arrhythmogenic cardiomyopathy (ACM) is a non-ischemic familial heart disease in which exercise constitutes a primary environmental risk. More specifically, in ACM subjects, physical training can trigger sudden cardiac death (SCD) and/or disease progression to heart failure (HF) [4,5,6]. ACM patients, like any other individuals, are also widely exposed to more elusive and chronic forms of environmental challenges, such as psychosocial stress (PSS).

PSS afflicts millions of people worldwide [7,8] and emanates from all the aversive social and psychological conditions that tax or exceed the physiological resources of an organism to cope with them. PSS can stem from a toxic work environment, social and financial difficulties, and socially restricting measures [3], such as those imposed by the COVID-19 pandemic [9]. Similar to other stressors, PSS impinges on the hypothalamus-pituitary-adrenal axis, leading to a surge of glucocorticoid hormones, such as corticosterone [10]. Short-term bursts of these hormonal adaptors help mammals cope with environmental stress; however, chronic activation of this pathway can negatively affect metabolism, reproductive cycle, cardiovascular function, and behavior and cognitive function [11]. Surprisingly, whether PSS acts as an external independent risk factor, and to what extent, for SCD or HF progression in subjects suffering from ACM is currently unknown.

Determining whether PSS is a risk factor in ACM is of paramount relevance for several reasons. First, ACM has a relatively high prevalence (1/5000 individuals) [12] and is a leading cause of SCD, particularly in the young and in athletes [13]. Second, physiological stress (e.g., exercise) increases disease penetrance, arrhythmic risk, and cardiac dysfunction in ACM [4,5,6]. Third, severe emotional distress/anxiety can contribute to Takotsubo cardiomyopathy, a non-ischemic acquired cardiac disorder sharing clinical features with ACM, such as ventricular dysfunction and life-threatening arrhythmias [14,15]. However, the association between PSS and cardiovascular conditions is extremely difficult to characterize and quantify. For example, when it comes to the specific case of heart disease, it is complicated by the necessity of scouting the contribution of external, environmental factors vs. cardiac disease-induced psychological discomfort. Against this background, we first tested whether there is any evidence suggesting a link between PSS levels, myocardial dysfunction, and arrhythmic burden in ACM patients. Furthermore, we then assessed whether PSS levels in ACM patients were associated with harboring a specific pathogenic desmosomal variant, and if so, does this association portend the severity of disease phenotypes. Then, we explored this correlative relationship via subjectification of PSS upon Desmoglein-2 mutant (*Dsg2*^mut/mut^) mice whose heart phenocopies human ACM using a universally accepted means to induce PSS in rodents, the resident-intruder (RI) paradigm test [16]. With this tool in hand, we sought to determine whether, in *Dsg2*^mut/mut^ mice, PSS can directly trigger sudden death and is associated with cardiac dysfunction progression and pathological remodeling.

## 2. Experimental Section

### 2.1. Animal Study Approval

All experiments conform to the Guide for the Care and Use of Laboratory Animals from the National Institutes of Health (NIH publication no. 85–23, revised 1996), and followed adherence to the Johns Hopkins School of Medicine Animal Care and Use Committee. Mice were given water and mouse chow ad libitum. A full genetic and phenotypic description of *Dsg2*^mut/mut^ mice used in these experiments can be found in Chelko et al. [17,18].

### 2.2. Human Study Approval

All participants met 2010 Task Force Criteria (TFC) for ACM [19], and participants provided written informed consent for participation in PSS questionnaires. This study was approved by the Johns Hopkins School of Medicine institutional review board (IRB Study No. NA_00041248; PI: Dr. Hugh Calkins). We administered PSS questionnaires to ACM patients during their most recent clinical follow-up to correlate PSS levels with each patient’s most recent functional phenotypes (i.e., Echo/MRI and 24 h Holter monitor/ZioPatch). Lastly, multivariate analysis was performed to assess whether a specific pathogenic desmosomal variant, or lack thereof, was associated with PSS scores and/or functional phenotypes. To perform such an analysis, gene variants were assigned the following numbers: (*n* = 0) for meeting TFC for ACM, but lacked a pathogenic variant; (*n* = 1) *PKP2* variant; (*n* = 2) *DSG2* variant; (*n* = 3) *DSC2* variant; and (*n* = 4) *DSP* variant. Number assignment was completely arbitrary, as confirmed by re-assignment of numbers, which had no impact on outcomes.

### 2.3. Perceived Stress Scale

The Perceived Stress Scale is one of the most extensively used psychometric tools for measuring the perception of stress in healthy and pathological populations [20,21,22]. It is a measure of the grade to which circumstances in one’s life are evaluated as stressful. The test also comprises queries about levels of experienced stress. The US population PSS average was obtained from S. Cohen and collaborators [23].

### 2.4. Animal Protocol

Fourteen-week old C57BL/6 wildtype (WT) controls and *Dsg2*^mut/mut^ mice were subjected to 14 days of resident-intruder (RI) paradigm to induce a condition of chronic psychosocial stress [2,24]. Levels of anxiety-like behavior were assessed via the light-dark box (LDB) test and cardiac function was assessed via echocardiography prior to and after 14 days of RI challenge. Post mortem, the levels of fibrosis and apoptosis were evaluated by Masson’s trichrome and Terminal deoxynucleotidyl transferase mediated dUTP Nick End Labeling assay (TUNEL), respectively.

### 2.5. Resident–Intruder Paradigm

We employed a modified version of the resident-intruder (RI) paradigm, as recently described [2]. The RI test, a psychosocial stress experimental model, induces offensive behavior (aggression of the resident mouse) and defensive/submissive behavior (avoidance and social defeat of the intruder mouse) [2,24]. Male CD1 mice, selected for their aggressive behavior, were placed individually in a single cage for seven days, allowing for the creation of an individual territory. Following seven days of habitat formation, each resident CD1 mouse received an intruder mouse (a WT control or a *Dsg2*^mut/mut^ mouse), and the two animals were allowed to interact freely for 10 min. After 10 min of animal-animal contact, mice were separated through a perforated Plexiglas partition halving the cage. This perforated partition maintained a continuous sensory contact (i.e., visual, auditory, and olfactory) for 24 h. The entire procedure was repeated every 24 h, for 14 consecutive days.

### 2.6. Light-Dark Box Test

The light-dark box (LDB) was used to measure anxiety-like behavior in mice [25], a test based on the contrast between the innate aversion for open illuminated areas and the spontaneous exploratory activity of rodents. The test apparatus consists of a box separated into a small (one third) dark chamber and a large (two thirds) brightly illuminated chamber. Mice were allowed to move freely between the two areas for 5 min, while their behavior was recorded by a camera. The total time spent in the lighted compartment is an index for bright-space anxiety, while the number of transitions between the two chambers is an index of activity-exploration because of habituation over time.

### 2.7. Cardiac Function and Circulating Corticosterone

Cardiac function was assessed in non-anesthetized mice via transthoracic echocardiography using a Vevo 2100 Visualsonic imaging system (FUJIFILM VisualSonics, Inc., Toronto, ON, Canada), as previously described [17,18]. Echocardiographic wall and chamber parameters and measurements were obtained using guidelines by the American Society of Echocardiography [26]. Three to five measurements per mouse were obtained and averaged, using short-axis, m-mode echocardiography, and parasternal long-axis view of the left ventricle (LV). Echocardiographs were recorded at baseline and following 14 days of RI test for each of the four groups. Following the collection of endpoint cardiac function (i.e., at 14 days of the protocol), blood was collected through the facial vein, then mice were euthanized and hearts excised to perform immunohistochemical experiments, as described below. Circulating corticosterone levels were determined by Enzyme-Linked Immune Assay (ELISA) (Corticosterone ELISA kit, Abcam, Cambridge, UK; Catalog no. ab108821) according to the manufacturer’s instructions.

### 2.8. Immunohistochemistry and Immunofluorescence

Myocardium was formalin-fixed and paraffin-embedded (FFPE), and 5 μm thick tissue sections were used for immunostains. TUNEL staining was used to detect myocardial apoptosis (In Situ cell death detection kit; Sigma, Saint Louis, MO, USA; Cat.No. 11684795910), as described previously [2,17]. Percent apoptotic cardiac cells were calculated via the number of TUNEL positive nuclei over the total number of cardiac nuclei per field (at a magnification of 100×, 3–5 confocal images taken per mouse and averaged). Additionally, myocardial fibrotic replacement was determined via Masson Trichrome (HT15 Trichrome Stain kit; Sigma) immunostain. Percent myocardial fibrosis was calculated as the sum of all fibrotic areas divided by total ventricular myocardial area using ImageJ version 1.53e software.

## 3. Results

### 3.1. Perceived Stress Levels in ACM Patients Correlate with Disease Severity

Previous studies showed heightened anxiety in ACM patients with an implantable cardioverter defibrillator (ICD) [27]; however, no studies have yet correlated the levels of perceived stress with the extent of cardiac phenotypes in patients with ACM. To fill this gap in our knowledge, we administered a psychometric test [20] to assess perceived stress levels in ACM patients during clinical follow-up appointments and correlated their most updated clinical phenotypes with their psychometric test outcomes. The study population consisted of forty subjects meeting 2010 Task Force Criteria (TFC) for ACM [19], which comprised of primarily probands (*n* = 33/40, 82.5%), an equal distribution of sexes (*n* = 18 males, *n* = 22 females; 45:55 ratio), and an average age at follow-up of 39 ± 12 yrs (mean ± StDev). Even after exhaustive genetic screening, nine patients harbored no known pathogenic variant implicated in ACM (*n* = 9/40; 22.5%), while the majority harbored pathogenic variants in desmosomal genes (*n* = 31/40; 77.5%); consisting of Plakophilin-2 (*PKP2*, *n* = 22/31; 71%), Desmocollin-2 (*DSC2*, *n* = 2/31; 6%); Desmoglein-2 (*DSG2*, *n* = 4/31; 13%) and Desmoplakin (*DSP*, *n* = 3/31; 10%).

Collectively, PSS scores obtained from ACM patients were similar to those found in the general US population [23] (mean ± StDev: 15 ± 7 vs. 13 ± 6, respectively; Figure 1a) and classified as low-to-moderate stress [23]. This evidence suggests that a cardiac disease condition may not, per se, lead to heightened levels of perceived stress in ACM patients. Intriguingly, a more comprehensive analysis revealed a robust correlation between PSS scores and clinical phenotypes. Specifically, percent right ventricular fractional area change (%RV FAC; *r* = −0.52, *p* < 0.01; Figure 1b), percent left ventricular ejection fraction (%LVEF; *r* = −0.41, *p* = 0.01; Figure 1c), and maximum premature ventricular contractions and supraventricular ectopics per 24 hrs (Max PVC; *r* = 0.33, *p* < 0.05; and Max SVE; *r* = 0.50, *p* < 0.01; Figure 1d–f) all indicated correlations between severity of cardiac dysfunction and PSS levels.

That said, these correlative outcomes were obtained in our collective ACM patient cohort, and thus, we sought to ascertain if these findings held true between ACM patients with a pathogenic desmosomal variant (i.e., genotype-positive (G+)) versus those who met TFC for ACM (i.e., phenotype positive (P+)), yet harbored no known pathogenic variant (i.e., gene-elusive patients). It should be noted that all ACM patients met clinical TFC for disease, regardless of variant status. Therefore, we reanalyzed our ACM patient PSS questionnaires by genetic variant (i.e., G+, P+ patients) and gene-elusive patients (i.e., G−, P+). However, no significant differences were found between PSS levels in TFC (G+, P+) and TFC (G−, P+) ACM patients (Figure 2a). Even though ACM patients indicated similar PSS levels between cohorts, we further probed our data set and performed a multivariate analysis assessing pathogenic variant against PSS scores and functional phenotypes (Figure 2b). At first glance, no such associations could be made, except an apparent trend between the presence of a pathogenic variant and %LVEF (Figure 2b, *p* = 0.1333). Albeit no differences in %LVEF between cohorts (Figure 2c), color-coating each data point by variant indicated two specific cohorts that primarily fell within the 95% confidence interval: (a) *PKP2* patients and (b) gene-elusive patients (Figure 2d). However, only ACM patients with a *PKP2* variant showed a strong correlation with increased PSS scores and reduced %LVEF (Figure 2e,f). No such correlation was apparent in *DSG2* (*r* = 0.32; *p* > 0.999), *DSP* (*r* = 0.0; *p* > 0.999) and *DSC2* (underpowered) cohorts. While findings in the *PKP2* patient cohort are certainly intriguing, caution is noted given findings are underpowered for *DSG2* (*n* = 4), *DSC2* (*n* = 2), and *DSP* (*n* = 3) cohorts.

Given the high prevalence of ICDs in our collective ACM cohort (*n* = 34/40; 85%), we could not analyze %RVEF due to an inability to perform cardiac magnetic resonance imaging in these individuals. Additionally, the lack of adequate numbers for three TFC (G+, P+) ACM patient cohorts is a limitation worth noting. Specifically, whether ACM patients harboring a *PKP2* variant are truly the only TFC (G+, P+) ACM patients with a correlative relationship between PSS levels and %LVEF is too early to call. However, when this dreadful disease is assessed collectively, our current findings in humans suggest a link between PSS levels and cardiac phenotypes in patients with ACM; yet, they do not directly probe the directionality of this correlative relationship.

### 3.2. The Resident-Intruder Paradigm Effectively Models PSS in Mice

In the human model, investigating a causal and not merely correlational role of PSS in the course of ACM disease is extremely challenging due to the multidimensional nature of this construct and to the ethical limitations that prevent any direct administration of stress to cardiomyopathic patients. To address this conundrum, we employed an animal model of PSS. The RI paradigm mimics the dynamics of territorial dominance and social hierarchy between a dominant male (i.e., the resident) and another male (i.e., the intruder); the latter is subjected to the territorial harassment and aggressive bullying of the former (Figure 3a). To quantify PSS-induced by this behavioral paradigm, we measured anxiety levels in 14-week old male wildtype (WT) and Desmoglein-2 mutant (*Dsg2*^mut/mut^) mice, a robust mouse model of ACM [17,18], using the light-dark box (LDB) test over the course of two weeks (Figure 3b). First, we noticed no differences in the LDB outcomes between cohorts in the absence of RI, as shown by the test’s two main parameters, i.e., time-in-light (Figure 3c) and the number of transitions (Figure 3d). These findings suggest that harboring a cardiac disease condition may not be incisive enough to induce high anxiety levels, at least in this ACM mouse model. Conversely, after 14 days of RI, both WT (as already shown in previous studies [28]) and *Dsg2*^mut/mut^ mice had significantly elevated anxiety-like behavior (Figure 3c,d). However, this phenomenon was substantially more pronounced in the ACM mutants. More specifically, we found that transitions from dark-to-light dropped more markedly in *Dsg2*^mut/mut^ vs. WT mice. To corroborate these behavioral findings, we measured the main hormonal marker linked to anxiety and chronic stress, corticosterone [29]. In the absence of RI paradigm, both WT and *Dsg2*^mut/mut^ mice had significantly lower corticosterone levels compared to PSS-challenged counterparts (Figure 3e). These data highlights two important findings: (a) harboring a deleterious cardiac substrate (i.e., desmosomal variant) does not appear to induce basal anxious behavior, and (b) that the RI paradigm is an effective PSS test to induce both behavioral signs of anxiety (Figure 3c,d) and a physiological stress response (Figure 3e), even in WT mice, as previously reported [28].

### 3.3. PSS Increases Mortality and Potentiates Myocardial Dysfunction in ACM Mice

After confirming that the RI test induces a state of PSS-related anxiety in both WT and *Dsg2*^mut/mut^ mice, we proceeded to analyze this impact on cardiac phenotypes, thus addressing this research’s central inquiry. First, we found that no deaths occurred in WT and *Dsg2*^mut/mut^ mice that were not exposed to the RI paradigm during the 2 weeks’ time of the protocol (Figure 4a; 100% survival for both). Conversely, while all WT mice survived the 14 days of RI challenge (100% survival), this was in stark contrast to PSS-challenged *Dsg2*^mut/mut^ mice that exhibited a 30% in-trial mortality rate (Figure 4a; 70% survival). Moreover, of the 70% (*n* = 7/10) of *Dsg2*^mut/mut^ mice that survived the RI challenge, these mutants showed a marked drop in both systolic and diastolic myocardial performance, compared to no sizable reduction in non-stressed WT and *Dsg2*^mut/mut^ mice or PSS-challenged WT mice (Figure 4b–f). In detail, both left ventricular ejection fraction (LVEF%) and fractional shortening (LVFS%) did not significantly change in WT mice at baseline and following the RI challenge, the same was true for non-stressed *Dsg2*^mut/mut^ mice. Conversely, mutant mice subjected to RI showed a strongly significant reduction (−25% baseline vs. 14 days, *p* < 0.001, Figure 4c,d) in both LVEF% and LVFS%.

In the same vein, a slight impairment in systolic function was observed in WT mice subjected to PSS stress, as assessed by echocardiographic LV internal diameter at systole (LVIDs; +4% increase) and end-systolic volume (ESV, +10% increase; Figure 4e,f). However, these PSS-induced changes in systolic function were not statistically significant compared to non-stressed WT mice. In complete contrast, *Dsg2*^mut/mut^ mice subjected to 14 days of PSS showed a substantial rise in LVIDs (+30% increase; Figure 4e) and ESV (+75% increase; Figure 4f). That said, PSS impaired ventricular relaxation in both WT and *Dsg2*^mut/mut^ mice, assessed via LVID at diastole (LVIDd; Figure 4g), albeit minimal in WT mice. However, diastolic dysfunction assessed by end-diastolic volume (EDV) was only observed in *Dsg2*^mut/mut^ mice subjected to PSS (+50% increase; Figure 4h). It should be noted that even without an environmental substrate (i.e., at baseline), *Dsg2*^mut/mut^ mice already display cardiac dysfunction compared to WT mice, as previously indicated for these mice at this age [17,18]. Yet, this set of data attests, for the first time that PSS can act as a potent (i.e., 2 weeks of RI) external stimuli contributing to sudden death and a primary aggravator of myocardial systolic and diastolic function in ACM-like subjects.

### 3.4. PSS-Induced Anxiety Correlates with Cardiac Dysfunction in ACM Mice

To further prove the point that a primary causal relationship exists between PSS-induced anxiety and the extent of cardiac dysfunction, we next correlated time-in-light (i.e., the most weight-carrying parameter of the LDB anxiety test [25]) with cardiac function (LVEF%). We found no such correlation exists in unstressed *Dsg2*^mut/mut^ and WT mice (Figure 5a,b) or PSS-challenged WT mice (Figure 5c). Conversely, PSS exacted a substantial toll in ACM mutants, which displayed a strong correlative relationship (*r* = 0.83, *p* < 0.05) between these two parameters (Figure 5d). These data validate the interpretation that ACM hearts harbor an ideal pathological substrate for which PSS can worsen ventricular function.

### 3.5. PSS Worsens Cardiac Fibrosis and Myocardial Apoptosis in ACM Mice

Fibrotic replacement of myocardium is a pathological hallmark of ACM [30], and glucocorticoids can stimulate fibrotic deposition via the mineralocorticoid receptor [31], particularly in the presence of heightened oxidative stress [31]. Therefore, we tested whether PSS is penetrant enough to induce myocyte loss and fibrotic replacement. First, we confirmed that *Dsg2*^mut/mut^ mice showed increased myocardial fibrosis compared to WT counterparts at baseline (Figure 6a,b). Of relevance, we noted that PSS challenge markedly expanded the extent of fibrotic deposition in ACM mutants (Figure 6a,b), and increased myocardial apoptosis by three-fold (Figure 6c,d). These results indicate that PSS can markedly exacerbate the functional and structural cardiac status in ACM subjects. Of note, PSS played a causal role in the extent of fibrosis, as there was a strong correlation with anxiety levels measured through the LDB test (*r* = −0.92, *p* < 0.05; Figure 6e). There was also a robust correlation between myocyte apoptotic levels and anxiety-like behavior in ACM mice exposed to PSS (*r* = −0.93, *p* < 0.01; Figure 6f). Thus, PSS can directly aggravate myocyte cell loss, and therefore, fibrotic replacement in ACM hearts.

## 4. Discussion

Our study provides the first benchmark evidence of a causal relationship between PSS, increased risk of sudden death, cardiac dysfunction, and myocardial remodeling in a murine model of ACM. Of relevance, we gained these findings in the absence of any other comorbidities, such as obesity, hypertension, or diabetes, which could further aggravate ACM disease progression. Present evidence suggests involuntary, intense, and persistent PSS as a previously unrecognized external perpetrator of cardiac adversities in ACM subjects, possibly as penetrant as voluntary exercise in fueling ACM disease progression in this already vulnerable patient population [4,5,6].

As pointed out, similarly to what is found in patients with hypertrophic cardiomyopathy and long-QT syndrome, reaching a diagnosis of ACM imposes a psychological burden on patients and the weight of SCD and/or genetic transmission [32]. Another layer of stress is channeled by the implantation of a cardioverter defibrillator (ICD) needed for primary or secondary prevention of SCD [33]. Indeed, patients wearing ICDs often experience anxiety and/or depression because of device-acceptance and anticipation of potential shocks [33,34,35], and these psychological repercussions are worse in subjects with inherited diseases than in patients who suffer from cardiac ischemic or valvular disease [36]. James and colleagues’ study in adult ACM patients with ICDs revealed how this population is at elevated risk of anxiety and how young patients face higher challenges with device-acceptance [27].

In the present study, we added another layer of complexity to this scenario. Like any other individual, ACM patients are exposed to more subtle and, very often, persistent forms of stress, such as that perpetrated by all the social conditions (work-related, family related, economic, etc.) that may lead to an imbalance between the demands placed on the individual and their ability to manage them, i.e., allostatic load imposed by psychosocial stress. However, like the perception of carrying an inherited cardiac disease and the need for an ICD implantation, PSS is a significant propeller of anxiety and depression. Indeed, it is also not so surprising that PSS in rodents, modeled by the RI paradigm, heightens the sensitivity of the heart to pro-inflammatory disorders [37]. Moreover, PSS causes lethality (~30%) in a mouse model of Duchenne muscular dystrophy (DMD, mdx mice) [38]. Using this same paradigm, we recently showed that obese mice subjected to social defeat present impaired cardiac contraction and relaxation (measured with pressure-volume relationships, i.e., in a load-independent manner), prominent cardiac apoptosis/fibrosis, redox imbalance, and arterial remodeling [2]. Moreover, we observed a marked decline in the cardiac and hippocampal levels of brain-derived neurotrophic factor (BDNF) and its associated, specific receptor, tropomyosin receptor kinase B (TrkB) [2]. In the brain of obese mice, PSS not only reduces hippocampal volume but hampers neurogenesis, fueling astrogliosis [2]. Thus, there are multiple routes through which PSS can jeopardize myocardial (and central) function, either in the presence of an inherited cardiac disease, such as DMD, or acquired morbidities or comorbidities, i.e., obesity. However, no previous study has ever addressed whether PSS has the potential to increase disease penetrance/expressivity in ACM-like subjects.

It is well established that pathogenic *PKP2* variants contribute to the vast majority of desmosomal ACM cases (>60%), while the second most prevalent are *DSG2* variants (>10%) [19]. While it may appear as a limitation to use a mouse model with the second most common desmosomal variant as seen in the patient population, *Dsg2*^mut/mut^ mice were chosen due to their robust recapitulation of key ACM functional and pathological phenotypes, even at rest [17,18], but particularly so, due to their maladaptive propensity to external stimuli [17]. As such, we used the RI paradigm in *Dsg2*^mut/mut^ mice. The RI test has been extensively employed to study cardiac alterations in response to PSS in WT healthy rodents [39,40,41,42]. First, we show that despite the profound structural and functional cardiac alterations harbored, *Dsg2*^mut/mut^ mice behave and display anxiety levels similar to their WT counterparts when unstressed. Thus, confirming that it is possible to study the impact of purely psychosocial stress driven disease progression. ACM mutant mice died prematurely during the RI test (with a mortality rate of 30%), whereas no deaths occurred in WT counterparts subjected to RI paradigm. No deaths occurred in *Dsg2*^mut/mut^ mice that were merely subjected to the same passage of time as age-matched *Dsg2*^mut/mut^ mice subjected to PSS. Moreover, ACM mice that survived the PSS paradigm displayed markedly worse cardiac dysfunction and remodeling, with pronounced apoptosis and fibrosis. Of relevance, to demonstrate the causal link between PSS and cardiac dysfunction in this ACM model, the prominent drop in contractility, rise in diastolic relaxation, and the extent of fibrosis and apoptosis strongly correlated with anxiety levels induced by the RI paradigm; a correlation evident only in mutant mice subjected to RI test. In fact, as shown above, in healthy WT mice, PSS is not incisive enough to sizably affect heart function and structure, but when impinging on a vulnerable ACM substrate (i.e., murine *Dsg2* loss-of-function), it has devastating effects. This evidence allows us to draw a new conclusion: environmental factors, such as PSS, may exacerbate myocardial structural and functional deficits, increasing the ACM disease progression. When interpreted through the lens of human ACM, one practical implication suggested by our experimental evidence is that, like exercise, PSS/social defeat may trigger or exacerbate an ACM substrate—transitioning cardiac dysfunction from latent to manifest. If so, PSS should be considered a risk factor in ACM disease progression.

Concerning the possible mechanisms at play, we can only speculate, at the moment that the allostatic load represented by chronic exposure to stress hormones such as corticosteroids, has an even more devastating effect on a heart already made vulnerable by a pathogenic desmosomal variant. Chronic exposure to high glucocorticoid levels harms the cardiovascular system through actions primarily mediated by the mineralocorticoid receptors [43,44]. In the present case, despite a similar rise in RI-induced corticosterone levels in WT and ACM mice, this hormonal surge may be impinging on an already vulnerable ACM heart that may hasten disease progression. Undoubtedly, one of these pre-existing factors is represented by the elevated sympathetic tone present in ACM patients, leading to impairment of presynaptic catecholamine reuptake [45], and enhanced catecholamine catabolism that contributes to cardiac dysfunction and remodeling [46].

Present findings come with limitations. Some are unavoidable, such as the intrinsic flaw of modeling human disease in animals. For instance, we do not know how the extent of PSS detected via ACM patient PSS questionnaires strictly adheres to that elicited by two weeks of an involuntary and intense RI test. However, the latter is a validated and universally accepted tool that has allowed many investigators to elucidate the PSS contribution on central and peripheral diseases [2,24,38]. Some aspects require further investigation, such as how PSS-induced anxiety and/or humoral perturbations weaken the heart, rendering it more prone to structural and functional deterioration. In the same vein, more studies are warranted to understand how heart-borne functional perturbations negatively reverberate on central functions, such as cognitive capacities and mood control. However, the present study was not conceived to dig into such specific mechanisms but to provide the initial critical groundwork needed to uproot future molecular pathways.

## 5. Conclusions

Our study demonstrates that in mice mimicking the ACM phenotype, a causal relationship between sudden death, cardiac dysfunction, and myocardial remodeling exists with quantifiable PSS readouts. Present evidence suggests PSS as an independent risk factor that alone or combined with other cardiac-intrinsic or environmentally linked triggers, could precipitate sudden death or foment disease progression in patients affected by ACM and possibly other forms of cardiomyopathy, such hypertrophic cardiomyopathy or long-QT syndrome.

## Figures and Tables

**Figure 1 jcm-09-03804-f001:**
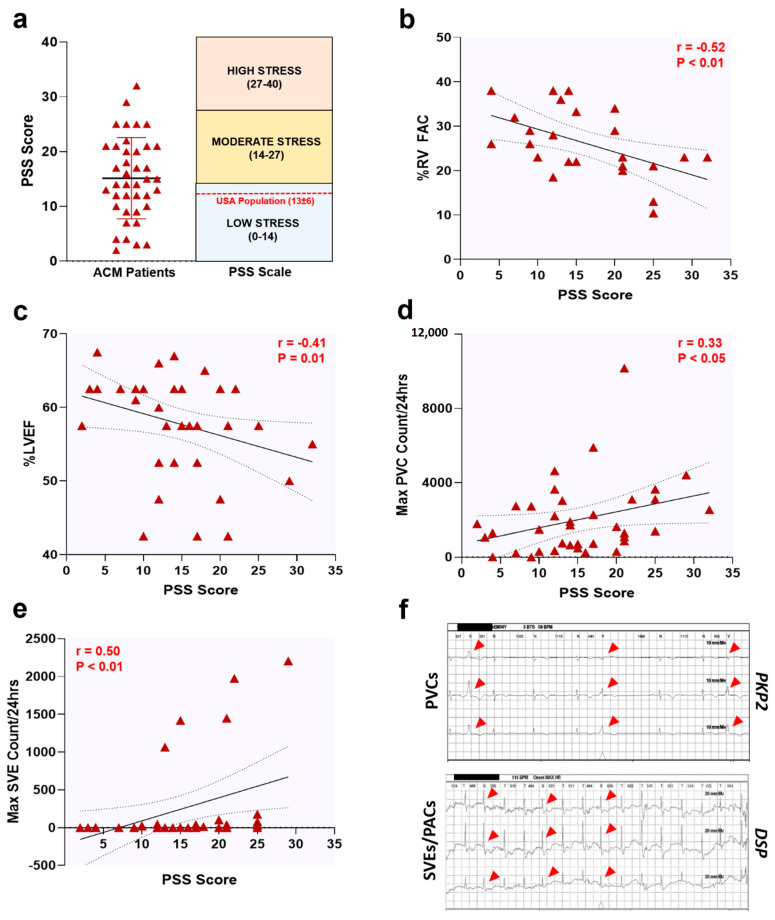
Perceived stress levels correlate with cardiac dysfunction in ACM patients. (**a**) PSS scores in *n* = 40 ACM patients. Data presented as Mean ± StDev. Upon further analysis, PSS scores correlated with ACM patient phenotypes; notably, (**b**) percent right ventricular fractional area change (%RV FAC), (**c**) percent left ventricular ejection fraction (%LVEF), (**d**) maximum premature ventricular contractions per 24 h (Max PVC Count/24 h), and (**e**) maximum supraventricular ectopics per 24 h (Max SVE Count/24 h). Data displayed in scatterplots; thick black line shows the best linear fit through the data; dotted lines represent two-tailed 95% confidence interval; text inset reports Spearman’s Rho value and associated *p*-value. For (**a**–**e**), each maroon triangle is an individual data point from a single patient. (**f**) Representative Holter strips showing PVCs (top panel) and SVEs (lower panel) from ACM patients harboring a Plakophilin-2 (*PKP2*) and Desmoplakin (*DSP*) pathogenic variant, respectively. Red arrowheads, PVCs and SVEs.

**Figure 2 jcm-09-03804-f002:**
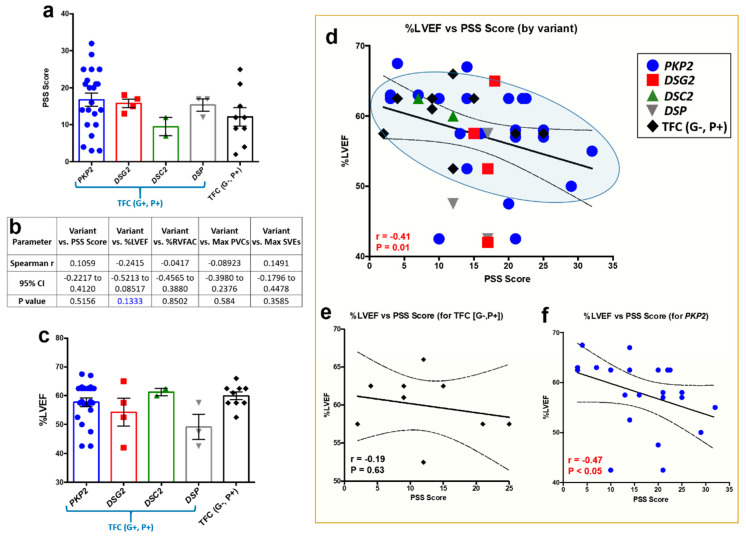
Multivariate analysis between perceived stress levels, pathogenic variant, and functional phenotypes in ACM patients. (**a**) PSS scores in *n* = 40 ACM patients, sorted by ACM patients with a pathogenic variant (i.e., TFC (G+, P+)) and gene-elusive patients (i.e., TFC (G−, P+)). No significance to note. (**b**) Multivariate analysis demonstrated a trend between presence of a pathogenic variant and %LVEF (noted in blue text). (**c**) Even though a trend was found between variant and %LVEF, no significant differences in %LVEF were found between cohorts. (**d**) However, color-coating each data point by variant or gene-elusive status indicated *PKP2* patients (blue circles) and TFC (G−, P+) patients (black diamonds) abundantly fell within the 95% confidence interval between PSS scores and %LVEF correlation analysis. (**e**,**f**) Yet, only *PKP2* patients showed a strong correlation between PSS scores and %LVEF. For (**a**,**c**), data presented as mean ± SEM. For (**d**–**f**), data displayed in scatterplots; thick black line shows the best linear fit through the data; dotted lines represent two-tailed 95% confidence interval; text inset reports Spearman’s Rho value and associated *p*-value.

**Figure 3 jcm-09-03804-f003:**
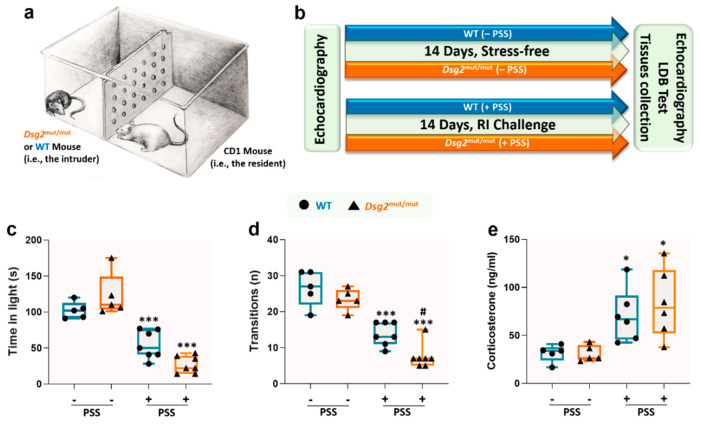
The resident-intruder paradigm induces anxiety behavior and a surge in corticosterone levels. Behavioral parameters assessed via the light-dark box (LDB) test and corticosterone levels from WT and *Dsg2*^mut/mut^ mice in presence or absence of resident-intruder (RI) paradigm. (**a**) Illustrative schematic depicting the RI paradigm test. (**b**) Graphical timeline for the experimental protocol. (**c**) Time spent in the light chamber during the LDB test. (**d**) Number of transitions between the two chambers during the LDB test. (**e**) Circulating corticosterone levels. Data presented as box-whisker plots, *n* = 5–7 mice/parameter/cohort. PSS, psychosocial stress. * *p* < 0.05, *** *p* < 0.001 WT (+PSS) vs. WT (–PSS) or *Dsg2*^mut/mut^ (+PSS) vs. *Dsg2*^mut/mut^ (–PSS); # *p* < 0.05 *Dsg2*^mut/mut^ (+PSS) vs. WT (+PSS) using One-Way ANOVA with Tukey’s post-hoc analysis.

**Figure 4 jcm-09-03804-f004:**
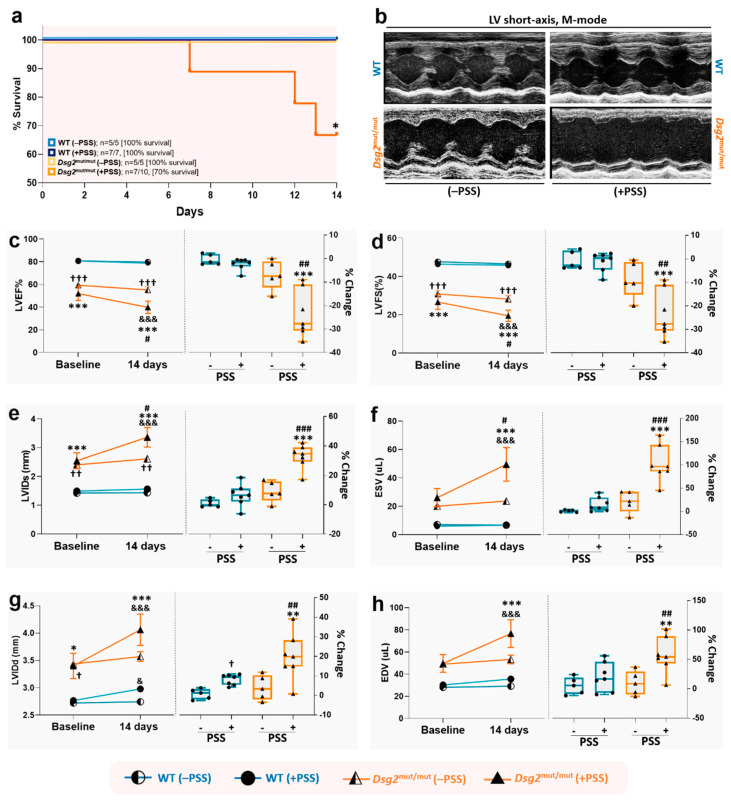
Psychosocial stress increases the risk of sudden death and worsens cardiac function in *Dsg2*^mut/mut^ mice. (**a**) Percent (%) survival curve in non-stressed (–PSS) and stress-induced (+PSS) WT and *Dsg2*^mut/mut^ mice. PSS, psychosocial stress. Data presented as % survival and n-values inset; * *p* < 0.05 for *Dsg2*^mut/mut^ (+PSS) vs. all cohorts via Wilcoxon survival test. (**b**) Representative left ventricular (LV) short-axis, m-mode echocardiography. Images representative of *n* = 5–7 mice/cohort. For **c**–**h**, the trend of echocardiographic parameters (y-axis) over time (x-axis) is shown on the left panels; and the percent changes (% Change) from baseline to the end of the protocol for each cohort, is shown on the right panels. (**c**) Percent Left Ventricular Ejection Fraction (LVEF%); (**d**) Percent Left Ventricular Fractional Shortening (LVFS%); (**e**) Left Ventricle Internal Diameter at systole (LVIDs; mm); (**f**) End Systolic Volume (ESV; µL); (**g**) Left Ventricle Internal Diameter at diastole (LVIDd; mm); and (**h**) End Diastolic Volume (EDV; µL). Data in left panels are presented as mean ± SEM; *n* = 5–7 mice/cohort/parameter; ^&^
*p* < 0.05; ^&&&^
*p* < 0.001 any cohort (at 14 days) vs. same cohort (at baseline) within cohort study, using two-way ANOVA with Sidak’s multiple comparisons post-hoc analysis; and * *p* < 0.01, *** *p* < 0.001 *Dsg2*^mut/mut^ (+PSS) vs. WT (+PSS) and WT (−PSS); ^#^
*p* < 0.05 *Dsg2*^mut/mut^ (+PSS) vs. *Dsg2*^mut/mut^ (−PSS), ^†^
*p* < 0.05, ^††^
*p* < 0.01, ^†††^
*p* < 0.001 *Dsg2*^mut/mut^ (−PSS) vs. WT (+PSS) and WT (−PSS), between cohort study, using two-way ANOVA with Tukey’s posthoc analysis. Data in right panels presented as box-whisker plots, *n* = 5–7 mice/cohort/parameter; ** *p* < 0.01, *** *p* < 0.001 *Dsg2*^mut/mut^ (+PSS) vs. WT (+PSS) and WT (−PSS); ^†^
*p* < 0.05 WT (+PSS) vs. WT (−PSS); ^##^
*p* < 0.01, ^###^
*p* < 0.001 *Dsg2*^mut/mut^ (+PSS) vs. *Dsg2*^mut/mut^ (−PSS), using one-way ANOVA with Tukey’s post-hoc analysis.

**Figure 5 jcm-09-03804-f005:**
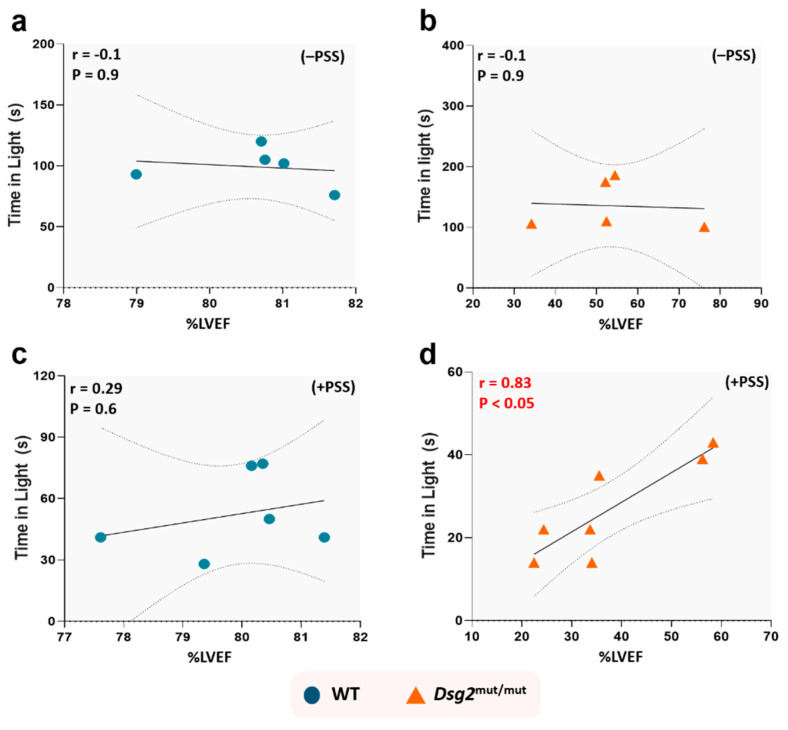
Anxiety levels induced by psychosocial stress correlate with cardiac dysfunction in *Dsg2*^mut/mut^ mice. No correlation was found between time spent in light with %LVEF from (**a**) WT mice and (**b**) *Dsg2*^mut/mut^ mice not subjected to RI (i.e., −PSS), or (**c**) WT mice after 14 days of RI challenge (i.e., +PSS). (**d**) Conversely, after 14 days of psychosocial stress, *Dsg2*^mut/mut^ mice showed a significant correlation between time spent in light and %LVEF (*r* = 0.83, *p* < 0.05). Data displayed in scatterplots; thick black lines represent the best linear fit through data; dotted lines represent two-tailed 95% confidence interval; text inset reports Spearman’s Rho values and associated *p*-values.

**Figure 6 jcm-09-03804-f006:**
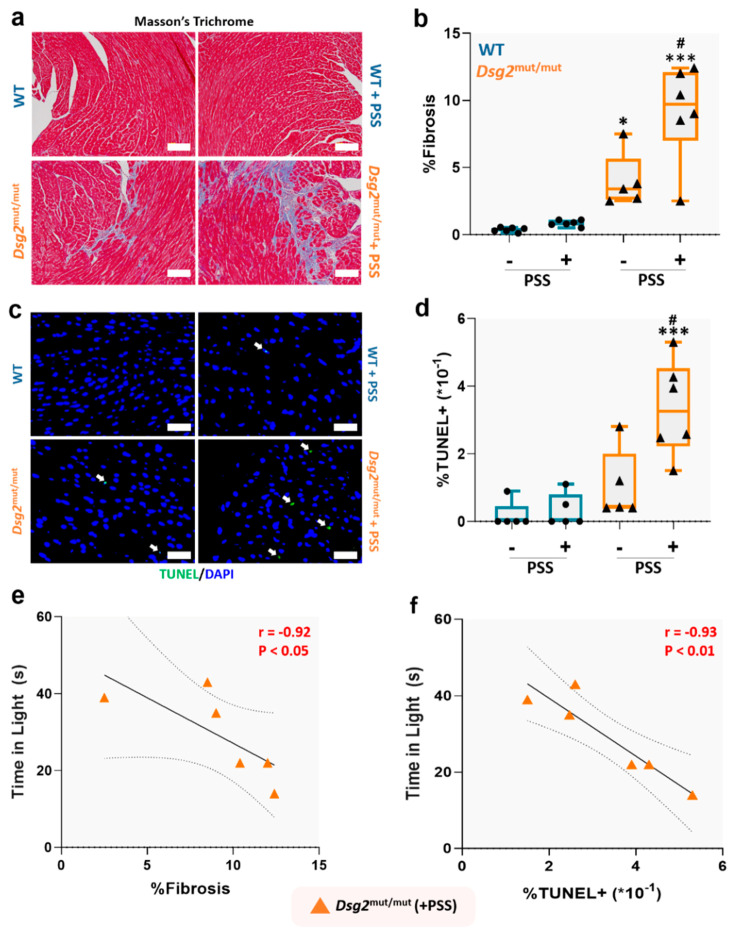
Psychosocial stress worsens pathological progression in *Dsg2*^mut/mut^ mice. (**a**) Representative images of Masson’s Trichrome immunostained myocardium. White scale bars, 100 µm. Images are representative of *n* = 5–6 mice/cohort; (**b**) Percent (%) myocardial fibrosis; (**c**) Representative images of myocardium immunostained for TUNEL and DAPI. Images are representative of *n* = 5–6 mice/cohort. White arrows, TUNEL positive (+) nuclei; white scale bars, 25 µm. (**d**) Percent (%) TUNEL+ nuclei in 5 microscopic fields from each mouse/cohort. For (**b**,**c**) data presented as box-whisker plots, *n* = 5–6 mice/cohort. PSS, psychosocial stress. * *p* < 0.05, *** *p* < 0.001 vs. WT (+PSS) and WT (−PSS); ^#^
*p* < 0.01 *Dsg2*^mut/mut^ (+PSS) vs. *Dsg2*^mut/mut^ (−PSS); using One-Way ANOVA and Tukey’s post-hoc analysis. For (**b**,**d**), black dots are WT mice (with or without PSS) and black triangles are *Dsg2*^mut/mut^ mice (with or without PSS). (**e**,**f**) Anxiety levels induced by PSS correlates with cardiac remodeling in *Dsg2*^mut/mut^ mice. Correlation between time spent in light and (**e**) percent (%) fibrosis and (**f**) percent (%) TUNEL+ nuclei in *Dsg2*^mut/mut^ (+PSS) mice. Data displayed as scatterplots; thick black lines show the best linear fit through data; dotted lines represent two-tailed 95% confidence interval; text inset reports Spearman’s Rho values and associated *p*-values.

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
