# Peer review of "Psychosocial Stress Hastens Disease Progression and Sudden Death in Mice with Arrhythmogenic Cardiomyopathy"

_jcm, 2020, doi:10.3390/jcm9123804_

Round 1

Reviewer 1 Report

The study by Agrimi et al gives new insights on the bilateral relationships between ACM and stress. The experiments are clear, well done and conclusions are very interesting. I have only minor comments and suggestions in Results and Discussion parts.

Results:

First Paragraph for each section explaining what will be described in results should be put in the Introduction.

  1. As it is already known that ACM is linked to stress, a new  clinical finding could have been to correlate genetic variants to level of stress.
  2. Fig 2, symbols in Legends are too small
  3. It seems that PSS doesn't induce an anxiety trait in WT. That is strange because it was found differently in many other studies using social defeat (in rats or mice), please explain and cite references in the Discussion (such as those from Sgoifo et al, De Boer et al, Sevoz-Couche et al, ...).
  4. The authors should explain why line 232, there is a mention of " mice that survive PSS..." , as we need to wait the discussion part to have the answer and the percent of mortality during PSS.
  5. Mention of the different baselines between WT and mutants in Fig 3 c-h is missing. The authors should explain why these cardiac parameters are not those obtained in patients. Triangles should be bigger.

Discussion:

  1. One limitation is that the well known animal model used here is a mutant for Desmoglein, but patients with this genetic variant represent only 13% here.
  2. I don't understand line 362 why the sentence begins with "Even more so"? this should be rephrased because it is not logical in the context of the preceding and following ideas.
  3. Again some references are missing because it was already shown that stress due to RI situation induces myocardic alteration (see in particular Costoli et al 2004, Carnevali et al 2013,  and more recently, Morais-Silva et al 2019, Brouillard et al 2020)

Author Response

Please see attached "Cover Letter and Response to Reviewers" document.

Reviewer 2 Report

This is very impressive translational research. The authors addressed the issue of how stress can precipitate sudden cardiac death or progression in heart failure patients with arrhythmogenic cardiomyopathy (ACM). They found stress correlated with the extent of arrhythmias and cardiac dysfunction. They then assessed the effects of stress, via the  resident-intruder (RI) paradigm, on Desmoglein-2 mutant mice, an animal model  of ACM. ACM mice succumbed to RI stress, while wild type mice did not. Desmoglein-2 mice that survived RI exhibited apoptosis and fibrosis. Sressed Desmoglein-2 mice also exhibited heightened anxiety.

I see no flaws with the work. My only suggestion is that with regard to potential mechanisms, the authors should consider the influence of the sympathetic nervous system, e.g., epinephrine, in contributing to ACM pathology.

Author Response

(The authors gave the same response as above.)
